# Highlighting the Role of Dielectric Thickness and Surface Topography on Electrospreading Dynamics

**DOI:** 10.3390/mi10020093

**Published:** 2019-01-28

**Authors:** Nikolaos T. Chamakos, Dionysios G. Sema, Athanasios G. Papathanasiou

**Affiliations:** School of Chemical Engineering, National Technical University of Athens, 15780 Athens, Greece; nhamakos@mail.ntua.gr (N.T.C.); dionissemas@gmail.com (D.G.S.)

**Keywords:** electrowetting, power-law exponent, structured dielectric

## Abstract

The electrospreading behavior of a liquid drop on a solid surface is of fundamental interest in many technological processes. Here we study the effect of the solid topography as well as the dielectric thickness on the dynamics of electrostatically-induced spreading by performing experiments and simulations. In particular, we use an efficient continuum-level modeling approach which accounts for the solid substrate and the electric field distribution coupled with the liquid interfacial shape. Although spreading dynamics depend on the solid surface topography, when voltage is applied electrospreading is independent of the geometric details of the substrate but highly depends on the solid dielectric thickness. In particular, electrospreading dynamics are accelerated with thicker dielectrics. The latter comes to be added to our recent work by Kavousanakis et al., *Langmuir*, 2018, which also highlights the key role of the dielectric thickness on electrowetting-related phenomena.

## 1. Introduction

Capillary-driven spreading of a droplet on solid substrate is related to many technological processes such as printing [1], coating and adhesion [2]. The prediction of spreading dynamics is however a tedious task considering that the dominating forces vary with time. In particular, early spreading dynamics are inertia dominated whereas viscosity is the dominating factor in late dynamics. The transition between inertial and viscous spreading has been related to a characteristic inertial time $\tau_{I}={\left(\rho R_{0}^{3}/\gamma_{\mathrm{LA}}\right)}^{1/2}$ where ρ is the fluid density, $R_{0}$ is the initial droplet radius and $\gamma_{\mathrm{LA}}$ is the liquid-ambient interfacial tension (e.g., $\tau_{I}=6$ ms for a glycerin droplet with $R_{0}=1.25$ mm). In addition, early-time spreading follows the power-law $r_{s}=Ct^{a}$ where $r_{s}$ is the spreading radius of the droplet (see also Figure 1), *C* and *a* is the prefactor and the exponent, respectively. We report that on completely wetting surfaces, *a* approaches the value of 0.5 [3,4]. Dynamic control of spreading behavior, by means of in situ controlling the parameters *a* and *C*, requires, however, the application of an external force. The above could be very useful for fabricating lab-on-a-chip devices where the dynamic control of droplet motion (e.g., spreading) is of utmost importance [5,6].

The early spreading dynamics can be manipulated by applying an external electric potential. In particular, it has been reported that electrostatically-induced spreading can increase the spreading exponent from *a* = 0.5 to *a* = 0.66 [7,8]. Thus, by applying a voltage between a droplet and an electrode under an insulating substrate, the wettability of the solid material is enhanced due to charge accumulation on the droplet surface (see also Figure 1). The above apparatus is similar to an electrowetting experiment [9,10]. Based on our recent work [11,12,13], where we outlined the importance of dielectric’s thickness and topography on electrowetting, some interesting questions have been raised: (a) What is the effect of the substrate’s structure on electrospreading behavior and (b) since the dielectric thickness affects the interface at the three-phase contact line, we ask whether it also influences the corresponding dynamics.

The scope of this article is to investigate the mechanism of electrostatically-assisted spreading by performing continuum-level simulations. The latter are not based on any simplifications regarding the actual shape of the droplet and the electric field distribution. In particular, we use a sharp-interface model, that has been developed by our team, where both the liquid-vapor and the liquid-solid interfaces are treated in a unified manner. Here we employ the notion of the Derjaguin (disjoining) pressure to model the micro-scale liquid-solid interactions [14,15,16]. In this work we investigate different substrate cases by varying the surface topography and the dielectric thickness. We are particularly interested in understanding the relation between the electrospreading dynamics and substrate characteristics.

We initially present the mathematical formulation used in the simulations and we demonstrate the experimental apparatus. Our numerical results are then validated against experimental measurements. Finally, we perform a computational analysis regarding electrospreading dynamics on various dielectric structures and thicknesses. Concluding annotations are made in the final section.

## 2. Materials and Methods

### 2.1. Electrospreading Modeling

In the performed simulations, we consider an axisymmetric droplet of a conductive liquid spreading on a structured insulating layer with thickness *d*, coating a base electrode, while subjected to an electric potential, *V*, (see Figure 1). The dynamics of the liquid droplet are governed by the Navier-Stokes equations i.e., the conservation of mass and momentum, given below:(1)$$\begin{array}{c} \rho \left(\frac{du}{dt}+u\cdot \nabla u\right)=-\nabla p^{L}+\mu \;\nabla^{2}u,\\ \nabla \cdot u=0, \end{array}$$
where, ρ is the fluid density, μ is the viscosity, $u=(u_{r},u_{z})$ is the fluid velocity field and $p^{L}$ the pressure. In our study, we have neglected any gravitational force in the Navier-Stokes equations. The tedious task of modeling multiple three-phase contact lines, on the structured surface, has been overcome by employing a sharp-interface scheme which has proven to be very efficient for the study of droplet’s dynamic behavior in our previous works [14,15,16]. According to this scheme, the liquid-vapor and the liquid-solid interfaces of the droplet are treated in a unified manner. Therefore, the solution of the Navier-Stokes equations (Equation (Equation 1)) is determined subject to a single stress balance boundary condition applied at the whole droplet surface (S), referred as the liquid-ambient interface from now on.

In particular, as thoroughly presented in [14,15,16], the liquid-solid interactions are lumped in an effective pressure term, $p^{\mathrm{LS}}$, which will now be accounted in the normal component of the interface force balance:(2)$$\tau_{\mathrm{nn}}{|}_{\mathrm{liquid}}=\Delta p-\gamma_{\mathrm{LA}}\;C-p^{\mathrm{LS}}-p_{\mathrm{el}},$$
where *C* is the local mean curvature, Δp is the pressure jump across the interface, $\gamma_{\mathrm{LA}}$ is the liquid-ambient interfacial tension, $p_{\mathrm{el}}$ is the electrostatic pressure generated as a result of the electric field. We note that since the liquid is considered as perfectly conducting (assuming a high electrolyte concentration), the electric field at the liquid surface is normal to the surface, and such is also the direction of the electrostatic pressure. In Equation (Equation 2) $\tau_{\mathrm{nn}}$ is the normal stress, defined as $\tau_{\mathrm{nn}}=n\cdot \tau \cdot n$ where τ is the viscous stress tensor ($\tau =\mu \left[\nabla u+{\left(\nabla u\right)}^{T}\right]$) and, n, the unit normal of the liquid-ambient interface (see Figure 1). The disjoining pressure, $p^{\mathrm{LS}}$, is commonly defined as the pressure in excess of the bulk pressure that is generated in a thin liquid film between two parallel plates. The latter can be attractive (negative disjoining pressure) or repulsive (positive disjoining pressure). The disjoining pressure in our case is formulated as follows:(3)$$\frac{R_{0}}{\gamma_{\mathrm{LA}}}\;p^{\mathrm{LS}}=w^{\mathrm{LS}}\left[{\left(\frac{\sigma }{\delta /R_{0}+\epsilon }\right)}^{C_{1}}-{\left(\frac{\sigma }{\delta /R_{0}+\epsilon }\right)}^{C_{2}}\right],$$

The above expression, introduced in [14,15,16], essentially approximates a Lennard-Jones type potential where $R_{0}$ is the initial droplet radius at *t* = 0. Apart from the latter expression, other possible formulations for the disjoining pressure (for example using an exponential function) could also be employed, as demonstrated in [17]. In Equation (Equation 3), the local minimum of the potential well is directly related to a dimensionless wetting parameter, $w^{\mathrm{LS}}$, which expresses the wettability of the solid material (an increase in $w^{\mathrm{LS}}$ would result in a greater depth of the potential, indicating more powerful liquid-solid attraction). Furthermore, the exponents $C_{1}$ and $C_{2}$ regulate the extend of the molecular interactions (large $C_{1}$ and $C_{2}$ decrease the area where these interactions are practically active). The gap, δ, between the liquid and the solid surface determines the nature of their interaction. The latter can be either attractive (modeling van der Waals interactions, for large δ) or repulsive (modeling steric forces and electrostatic interactions determined by an electric double layer overlap, for small δ) [18]. When a perfectly flat solid surface is considered, the gap, δ, is determined as the vertical distance between the liquid surface and the solid substrate. The definition of δ is, however, not a trivial task for non-flat, rough, solid surfaces. In this case, we define, δ, as the Euclidean distance from the solid boundary. This quantity is obtained by solving the Eikonal equation [19], which gives the shortest distance from a specified boundary (structured or even arbitrarily shaped). The current formulation thus requires that the liquid and the solid phases are always separated by an intermediate layer, with thickness $\delta_{\min}$, which is stabilized by the action of the disjoining pressure (see Figure 1). In particular, at $\delta =\delta_{\min}$ the repulsive and attractive forces balance each other. Thus, further reduction of the liquid-solid distance, below $\delta_{\min}$, would generate a substantial repulsion. The minimum allowed liquid-solid distance, $\delta_{\min}$, is determined by the constants σ and ϵ [14]. Specifically, for $\delta =\delta_{\min}\Leftrightarrow p^{\mathrm{LS}}=0\Rightarrow \delta_{\min}=R_{0}\left(\sigma -\epsilon \right)$ (see also Figure 2).

Considering the tangential stress component along the liquid surface, we employ a Navier slip model with an effective slip coefficient, $\beta_{\mathrm{eff}}$, active only in the vicinity of the solid surface:(4)$$\tau_{\mathrm{nt}}{|}_{\mathrm{liquid}}=\beta_{\mathrm{eff}}\;\left(t\cdot u\right),$$
where $\tau_{\mathrm{nt}}=n\cdot \tau \cdot t$ represents the shear stress and t is the unit tangent of the liquid-ambient interface (see Figure 1). We note that, a uniform interfacial tension along the interface has been considered in the above equation (i.e., $\nabla_{s}\gamma_{\mathrm{LA}}=0$). The Navier slip model is active only in the vicinity of the solid surface, and this is achieved by using an effective slip coefficient, $\beta_{\mathrm{eff}}$, of the following form:(5)$$\beta_{\mathrm{eff}}=\frac{\mu \;\beta_{\mathrm{LS}}}{R_{0}}\left(1-\tanh\left[p_{\mathrm{trs}}\left(\frac{\delta }{\delta_{\min}}-1\right)\right]\right).$$

In the above formulation, which is presented in [15], the dimensionless slip parameter, $\beta_{\mathrm{LS}}$ (representing a scaled inverse slip length), controls the adhesion strength of the liquid on the solid surface. The above formulation is a lucid way to express, in a continuous manner, the transition from a shear-free boundary condition, applied on the liquid-ambient interface, to a partial slip boundary condition along the liquid-solid interface. In particular, in the limit $\delta \approx \delta_{\min}$, the above equation reduces to $\beta_{\mathrm{eff}}=\mu \beta_{\mathrm{LS}}/R_{0}$, whereas for $\delta >\delta_{\min}$ yields $\beta_{\mathrm{eff}}=0$. In Equation (Equation 5) the parameter, $p_{\mathrm{trs}}$, provides a sharp, however, continuous and smooth, transition between these two regimes. We note that, in the computations presented in this work, we assume $p_{\mathrm{trs}}$ = 5. Typical values of the dimensionless slip parameter, $\beta_{\mathrm{LS}}$, are of the order of the scaled inverse minimum distance ($R_{0}/\delta_{\min}$), thus in this case $\beta_{\mathrm{LS}}$ = 10^3^.

In Equation (Equation 2) the electrostatic pressure term, given by $p_{\mathrm{el}}=\frac{\epsilon_{0}E^{2}}{2}$ where $\epsilon_{0}=8.854\times 10^{-12}$ F/m is the vacuum permittivity and *E* the magnitude of the electric field strength, incorporates the effect of the electric field, and acts at the liquid-ambinet interface, with a negative contribution to the total pressure [9]. The electric field strength, E=−∇ϕ, where ϕ is the electric potential, is calculated by solving the equations of electrostatics (Gauss’ law for electricity):(6)$$\nabla \cdot (\epsilon_{0}\epsilon_{r}\nabla \phi )=0,$$
for both the ambient phase and the dielectric material. We note that Equation (Equation 6) is not solved inside the droplet since a conductive liquid is considered. In the performed computations, the permittivity, $\epsilon_{r}$, is assumed to be given by a continuous function of this form, $\epsilon_{r}=\frac{\epsilon_{d}+\epsilon_{s}}{2}+\frac{\epsilon_{s}-\epsilon_{d}}{2}\tan h\left(\left(z-f_{\mathrm{solid}}\left(r\right)\right)\alpha_{\mathrm{trs}}\right)$, where $f_{\mathrm{solid}}$ is a function that descripes the solid topography. This expression ensures a continuous transition between the solid-ambient interface, i.e., the permittivity of the ambient phase and the dielectric material. In particular, the permittivity, $\epsilon_{r}$, becomes equal to $\epsilon_{s}$ in the ambient phase (insulating medium) whereas it equals to $\epsilon_{d}$ for the solid dielectric, respectively. In addition, when the parameter $\alpha_{\mathrm{trs}}$ attains a high value, a sharp transition between the two regions is achieved. For the simulations in this work, it is assumed that $\alpha_{\mathrm{trs}}$ = 500. Equation (Equation 6) is solved accounting for the following boundary conditions: ϕ=V, imposed at the liquid-ambient interface, where, *V*, is the voltage applied between the base electrode and the conductive droplet and, ϕ=0, applied at the bottom of the solid dielectric (base electrode) (see also Figure 1).

As a measure of the effect of the electric field we select the dimensionless electrowetting number, $\eta =\frac{\epsilon_{0}\epsilon_{r}V}{2d\gamma_{\mathrm{LA}}}$, scaling the strength of the electrostatic forces with the capillary forces in the system, under the assumption of a uniform electric field at the liquid-solid interface (for an ideal parallel plate capacitor). In the above equation, *d* denotes the dielectric thickness. For structured surfaces, *d*, is defined as the total thickness, i.e., $d=d_{\mathrm{bulk}}+d_{h}$, where $d_{\mathrm{bulk}}$ is the thickness of the bulk dielectric and $d_{h}$ the height of the dielectric protrusions (see Figure 1). Finally, we impose the following kinematic boundary condition along the droplet-ambient interface:(7)$$(u_{\mathrm{mesh}}-u)\cdot n=0,$$
where $u_{\mathrm{mesh}}$ is the velocity of the moving mesh at the interface. The model described above has been implemented in the commercial software package COMSOL Multiphysics^®^.

### 2.2. Electrospreading Experiments

To support our computational findings, we perform electrospreading experiments in a setup described in Figure 1. We have used a flat and smooth stack dielectric consisting of 400 nm tetraethoxysilane (TEOS) and 1 μm of poly- (methyl methacrylate) (PMMA). The dielectric is also coated by a thin layer of polytetrafluoroethylene (PTFE) in order to increase the Young’s contact angle (which quantifies the wettability of the substrate). In particular, the Young’s (or equilibrium) contact angle of a glycerin droplet, which is used in the following experiments, in the case where no voltage is applied, has been measured as $\theta_{Y}$ = 102${}^{\circ }$ by using the ramé-hart Model 590 goniometer/tensiometer. We also note that mineral salts have been added to the glycerin solution (99.5%, Sigma-Aldrich, St. Louis, MO, USA) for increasing its electrical conductivity.

The experiments are performed as follows: Initially, a voltage is applied between the base electrode (see Figure 1) and a stainless steel needle which is located approximately 2 mm above the solid substrate. A glycerin droplet is then generated at the end of the needle by using an auto dispensing system by ramé-hart. The volume of the droplet increases until touching and spreading on the solid substrate. The droplet spreading is captured by a high-speed camera (Olympus i-Speed 2), with 2000 frames per second. As a backlight source we used a high power OSTAR LED by OSRAM. Finally, the spreading radius of the droplet was measured using real-time image processing software that was developed in house.

## 3. Results and Discussion

Electrospreading experiments, as well as simulations, were performed on a flat and smooth dielectric surface using a conductive glycerin droplet (with ρ = 1261 kg/m^3^, μ = 1.41 Pa s and $\gamma_{\mathrm{LA}}$ = 0.063 N/m at 20 ${}^{\circ }$C [20]), and the results are presented in Figure 3. It is notable that the performed experiment follows the predictions for the spreading radius obtained by our computational model.

The experimental parameters, that have been used as inputs in the simulation, are: Young’s contact angle $\theta_{Y}$ = 102${}^{\circ }$ and electric potential *V* = 100 Volt. In addition, considering that there is an uncertainty in the permittivity values of the stack dielectric (due to the multiple layers of materials), we experimentally determine the capacitance per unit area, $c=\epsilon_{0}\epsilon_{d}/d$, by performing a static electrowetting experiment. In particular, the Lippmann equation [9]:(8)$$\cos\theta =\cos\theta_{Y}+\frac{cV^{2}}{2\gamma_{\mathrm{LA}}},$$
is fitted to the experimental measurements of the contact angle, θ, versus the applied voltage, *V*, for the corresponding dielectric substrate. Finally, we obtain: *c* = 1.2 × 10^−5^ F m^−2^ which is used in the simulations. This results in a electrowetting number $\eta =\frac{cV^{2}}{2\gamma_{\mathrm{LA}}}$ = 0.95. We also note that, following our previous studies, the disjoining pressure parameters we use in the computations are $C_{1}$ = 12, $C_{2}$ = 10, σ = 9 × 10^−3^, ϵ = 8 × 10^−3^ and $\beta_{\mathrm{LS}}$ = 10^3^. In particular, in [14,21] we have shown that our results converge to the experimental measurements as scaling down the range of liquid-solid interactions. The above set of parameters thus correspond to the maximum length scale of the liquid-solid interactions, below which our computational results remain practically unaltered.

Since we have performed a validation of our model, we will proceed to computationally examine the flow dynamics of a conductive glycerin droplet (with $R_{0}$ = 1.25 mm) spreading, under the effect of electric field, on various solid dielectrics. We consider three different dielectric cases: (a) flat, (b) arbitrary-structured and (c) stripe-structured. We also study various dielectric thickness cases ranging from *d* = 50 to 150 μm. We select thicknesses in this range because we recently found an important effect of dielectric thickness when electrowetting is performed in rough surfaces [11,12,13]. In the cases of a structured dielectric, the dielectric thickness, *d*, is considered as the total thickness i.e., the thickness of an effective layer that includes the surface topography features (see Figure 1). In all simulations that will be presented below the relative permittivity $\epsilon_{d}$ = 3.8 (we assume a silicon dioxide material), while $\epsilon_{s}$ = 1 (air ambient). In addition, we consider for the solid material a Young contact angle, $\theta_{Y}$ = 114${}^{\circ }$ (assuming a PTFE coating on the silicon dioxide substrate).

In Figure 2 we demonstrate two solid dielectric cases with (a) arbitrary-structured and (b) stripe-structured topography. We note that the dielectric thickness is equal in both cases (*d* = 50 μm with $d_{h}$ = 20 μm). Although different wetting states are observed in these two cases (Wenzel state at the arbitrary-structured topography and Cassie-Baxter state at the stripe-structured topography) it is notable that the spreading radius is virtually the same at *t* = 5 ms (see the insets at the right side of Figure 2).

To further investigate the effect of solid geometry on electrospreading dynamics, in Figure 4 we present the temporal evolution of the contact radius of a glycerin droplet, spreading under the effect of electric field, for a flat, an arbitrary-structured and a stripe-structured solid topography (the surface topographies are presented in Figure 2). Starting with the case where no voltage is applied (Figure 4a) we observe that the spreading dynamics are highly influenced by the solid geometry, especially for *t* > 2 ms. In particular, the spreading radius on the structured dielectrics deviates (i.e., moves slower) from the one of the flat substrate. The fluctuations observed in the spreading radius are attributed to the non smooth advancing of the liquid interface front over the solid surface protrusions of the structured surfaces. It is interesting, though, that the spreading curves approach each other by increasing the applied voltage (η = 0.25 to 0.48 in Figure 4b,c). Essentially, the electrostatic force highly accelerates the droplet advancing front, nearly annihilating the influence of the substrate topography. We have calculated the relative strength, $\lambda_{F}$, defined as the ratio of the electrostatic pressure term, $p_{\mathrm{el}}$, over the disjoining pressure, $p^{\mathrm{LS}}$, in the vicinity of the three-phase contact line [12]. Results show that as the electrowetting number increases the relative strength of the electrostatic pressure term also increases. In particular, for η= 0.25, 0.48 and 0.75 the $\lambda_{F}$ becomes 0.23, 0.52 and 1.11 respectively (for *d* = 50 μm). This indicates that the electric field becomes the dominant factor in electrospreading dynamics as the applied voltage is increased.

Since the electric field strongly affects the spreading dynamics, its distribution, which in turn governs the electric stress distribution, at the liquid interface should strongly affect the dynamics, too. We have recently found that both the field distribution and the liquid interfacial tension shape at the three-phase contact line, is substantially affected by the dielectric thickness, especially for thick dielectrics as a result of fringe field contributions (see [11,12,13]). Here we study how dielectric thickness affects electrospreading. In particular, in Figure 5 we have evaluated the spreading exponent of the power-law, as fitted on the corresponding results obtained by simulations, for flat and smooth solid substrates of thickness *d* = 50, 100 and 150 μm. Previous studies have reported that the power-law exponent increases as the Young’s contact angle decreases [4]. Such an argument is also observed in our simulations since the exponent increases by increasing the electrowetting number (the contact angle decreases due to electrowetting action). Surprisingly enough, we also observe that the power-law exponent depends on the thickness of the solid dielectric. In particular, it considerably increases with the dielectric thickness. This behavior gets even more pronounced for higher applied voltages.

To highlight the effect of electric field distribution and the concomitant electric stress distribution at the liquid surface, we evaluate the integral of the projection of the electrostatic pressure in the *r*-direction i.e., parallel to the spreading direction. This quantity, $F_{el_{r}}=2\;\pi \;\int_{s\;=\;0}^{s\;=\;s_{\max}}\;p_{\mathrm{el}}\;r\;n\cdot r\;ds$, can be considered as an effective driving force of spreading, where, r, is the unit vector in the radial direction and, *s*, is the arc-length of the effectively one dimensional droplet surface (see Figure 1). Its evolution in time is shown in Figure 6. An effective impulse can be evaluated by integrating the effective force over time: $J=\int_{t\;=\;0}^{t\;=\;5\mathrm{ms}}\;F_{el_{r}}\;dt$. The integrals yield: ${J|}_{d\;=\;50\mu m}$ = 5.2 × 10^−7^ N s whereas ${J|}_{d\;=\;150\mu m}$ = 7.0 × 10^−7^ N s for the thin and thick dielectric cases respectively. The above seem to indicate that, for the case of the thick solid dielectric (*d* = 150 μm), a higher net force (and impulse) is generated, thus resulting in an accelerated spreading behavior as demonstrated in Figure 5. This behavior, i.e., the significant dependence of electrospreading on the dielectric thickness, but its independence on the surface structure (at least for the structures studied), which according to our knowledge has not been reported in the literature before, could be potentially exploited for improving the electrostatic manipulation of droplets on lab-on-a-chip devices [5,6]. Examples of such devices, where accelerated droplet dynamics are desirable, are: electrowetting switches [22], electrowetting-based e-paper [23] as well as variable-focus liquid lenses [24].

## 4. Conclusions

In this work we investigated the impact of the solid topography and dielectric thickness on electrostatically-induced spreading. In particular, after validating our computational model with electrospreading experiments, we then focused on the geometric details of the solid dielectric. We concluded that the electrostatic pressure, when voltage is applied, downgrades the influence of the solid topography on the spreading dynamics. This phenomenon gets even more pronounced when applying higher voltages. We have also investigated the effect of the dielectric thickness on electrospreading, deducing that the dynamics are accelerated with thicker dielectrics.

The latter argument comes to be added to our previous works that highlights the role of dielectric thickness in reversible electrowetting [11,12,13]. In particular, in [11] we have experimentally showed that collapse, i.e., Cassie-Baxter to Wenzel, electrowetting transitions of a droplet on a superhydrophobic surface, can be prevented by using an adequately thick solid dielectric. Thus the thickness of the dielectric seems to play a role more important than argued in the literature up to now. Future work focuses on performing electrospreading simulations and experiments on superhydrophobic surfaces with various thicknesses for further investigating the interplay of topography and dielectric thickness.

## Figures and Tables

**Figure 1 micromachines-10-00093-f001:**
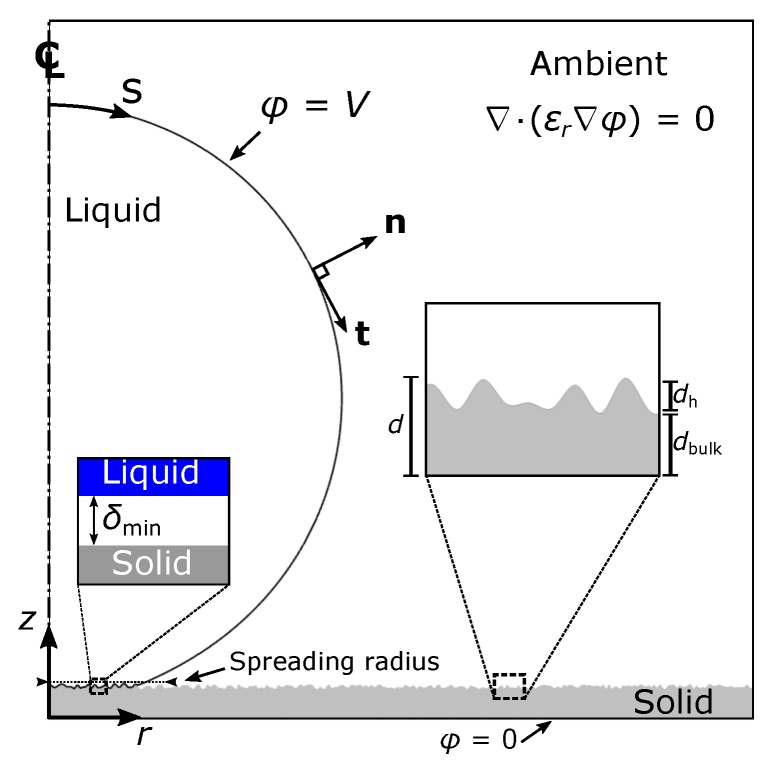
Schematic of the electrospreading setup of an axisymmetric droplet on a structured dielectric substrate (not drawn to scale).

**Figure 2 micromachines-10-00093-f002:**
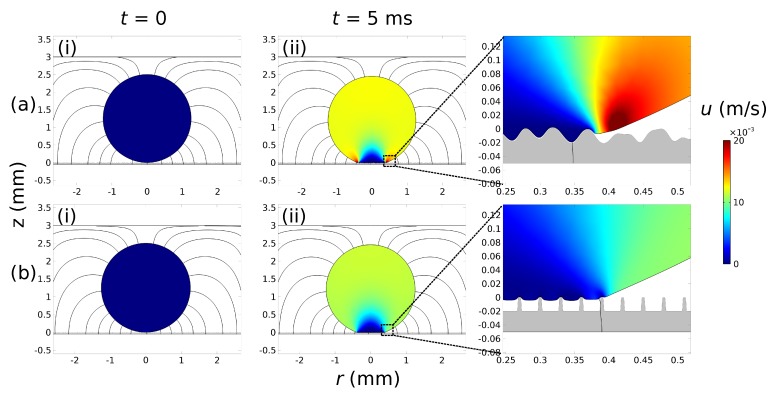
Visualization of the velocity magnitude (at *t* = 0 and *t* = 5 ms) of a glycerin droplet on two different structured solid dielectrics: (**a**) arbitrary-structured and (**b**) stripe-structured ($\theta_{Y}$ = 114°, $d_{h}$ = 20 μm and *d* = 50 μm for both solid dielectric cases). The electric field lines are also depicted (a voltage of 220 V, or η = 0.25, is applied in both cases). As observed in the inset, a Wenzel state is observed at the arbitrary-structured topography (**a**) whereas a Cassie-Baxter state is accommodated in stripe-structured topography (**b**). The disjoining pressure parameters we use are, according our previous work (see [14,15,16]): $C_{1}$ = 12, $C_{2}$ = 10, σ = 9 × 10^−3^ and ϵ = 8 × 10^−3^ (resulting in $\delta_{\min}$ = 1.25 μm) while the dimensionless slip parameter: $\beta_{\mathrm{LS}}$ = 10^3^.

**Figure 3 micromachines-10-00093-f003:**
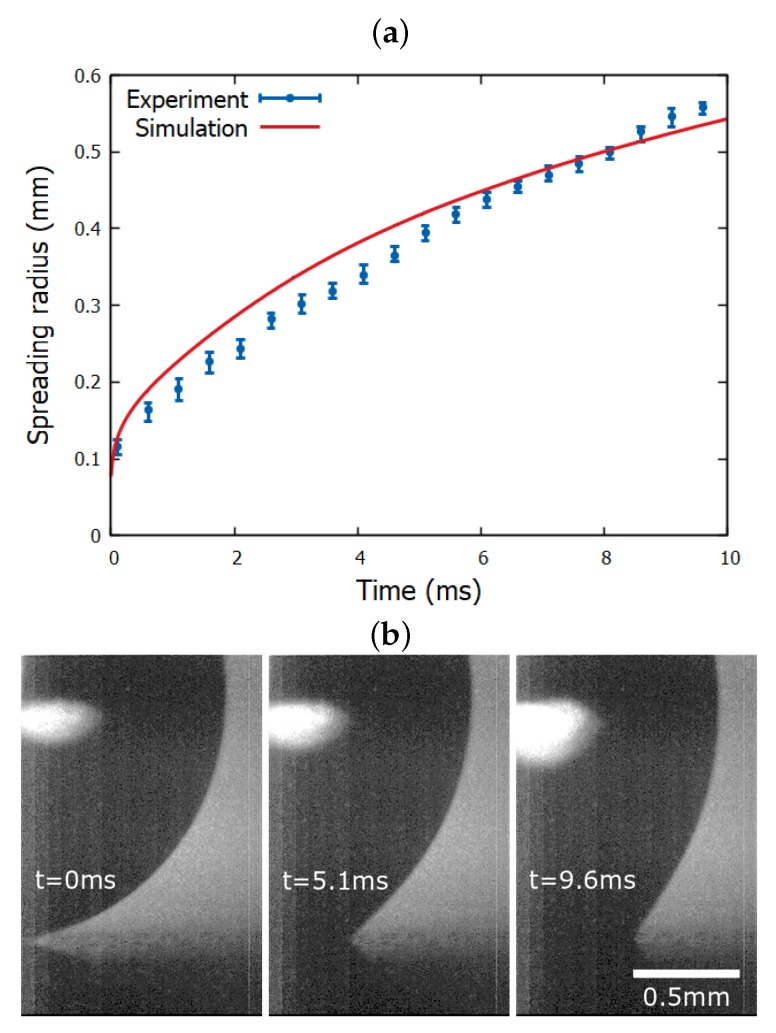
(**a**) Temporal evolution of the contact radius of a glycerin droplet as obtained from experiment and a corresponding simulation for electric potential ϕ = 100 V (η = 0.95) on a flat and smooth solid dielectric. (**b**) Snapshots of the spreading droplet at *t* = 0, 5.1 and 9.6 ms, under the effect of electric field, as captured by a high-speed camera.

**Figure 4 micromachines-10-00093-f004:**
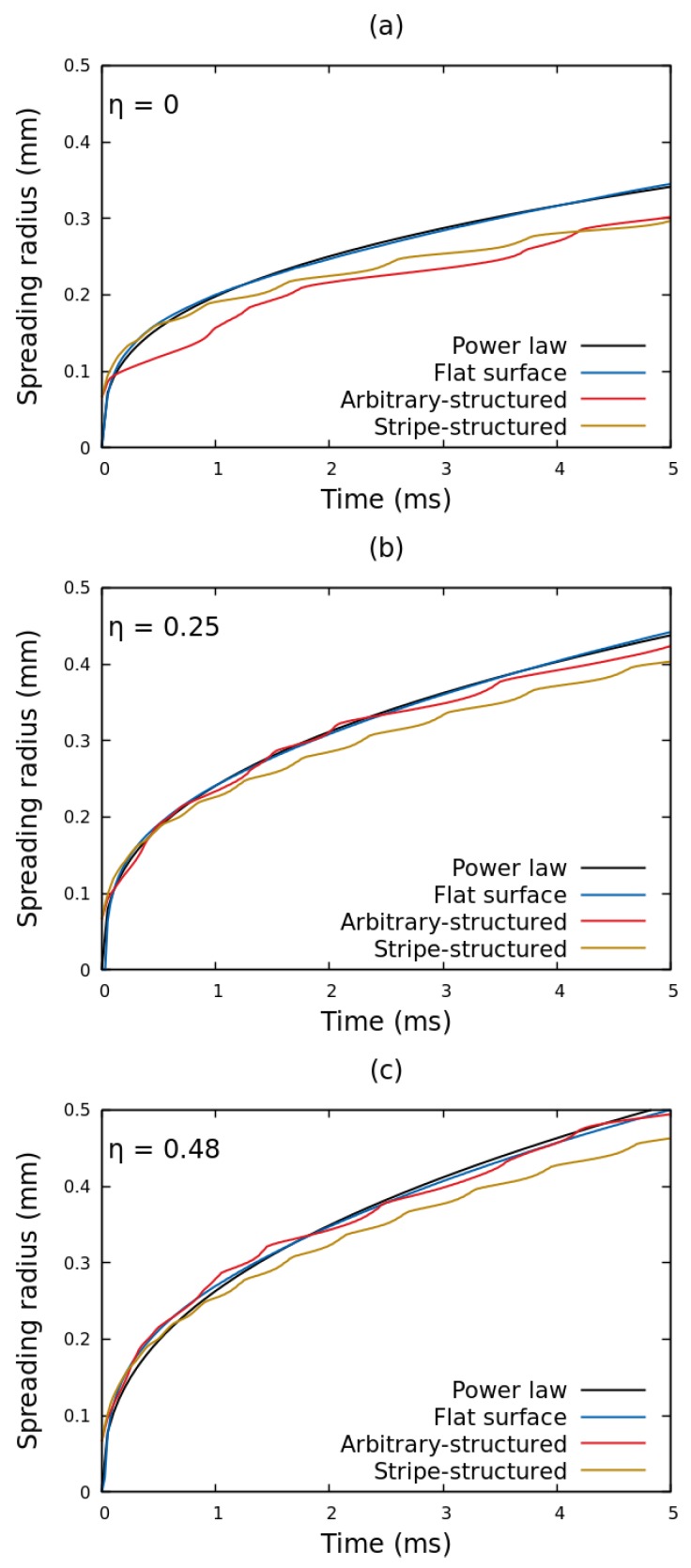
Temporal evolution of the contact radius of a glycerin droplet, spreading under the effect of electric field, for various solid dielectrics and electrowetting number, η, cases: (**a**) η = 0, (**b**) η = 0.25 and (**c**) η = 0.48. A power-law, $r_{s}=Ct^{a}$, which describes the spreading dynamics, is also visualized. The power-law exponents, *a*, fitted on the flat surface case are: (**a**) 0.34, (**b**) 0.37 and (**c**) 0.41. The dielectric thickness, *d*, is set to 50 μm for all cases.

**Figure 5 micromachines-10-00093-f005:**
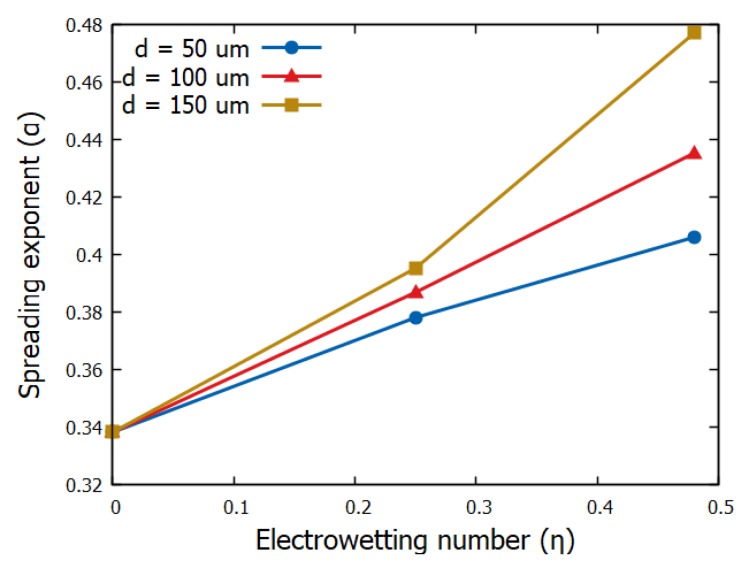
Spreading exponent, *a*, of the power-law, $r_{s}=Ct^{a}$, fitted on the results obtained by simulations for flat and smooth solid substrates with various thicknesses.

**Figure 6 micromachines-10-00093-f006:**
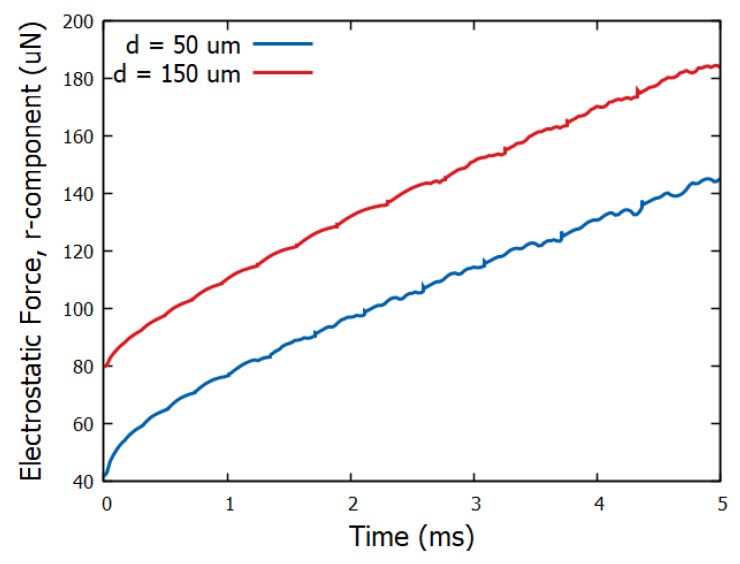
Effective electrostatic force in the radial direction, $F_{el_{r}}$, induced by the different electric field distributions for the thin (*d* = 50 μm) and the thick (*d* = 150 μm) dielectric cases (η = 0.48).

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
