# Peer review of "Highlighting the Role of Dielectric Thickness and Surface Topography on Electrospreading Dynamics"

_micromachines, 2019, doi:10.3390/mi10020093_

Round 1

Reviewer 1 Report

The article is a well-written description of an experimentally validated model of droplet spreading dynamics under the influence of an electric field and on patterned and smooth substrates. The conclusions are interesting and tie well into other published works in this field. Specific comments follow.

1.  Parenthesis missing in line 82

2.  More explanation is needed for sigma and epsilon are in equation 3 except that they are constants as mentioned on line 95. What are their physical meanings and values?

3.  Are equations 3 and 5 taken from a reference or were they formulated for this work? If from a reference, the authors should consider including it. If formulated for this paper, additional wording should be included.

4.  Justification should be added regarding the assumed values for p_trs and alpha_trs on lines 108 and 124, respectively.

5.  Please specify if the values for glycerin density, viscosity, and surface tension were measured or taken from a reference.

6.  The authors should consider expanding on their statement that the results of this study can be used for “improving the electrostatic manipulation of droplets on lab-on-a-chip devices.” Furthermore, the paper could be strengthened by adding more examples of how these findings might be important. If sufficient examples were added, the Results section could be renamed Results and Discussion (see comment # 7).

7.  The Discussion section is more a Conclusion and should therefore be renamed.

Author Response

We have uploaded our response in the PDF file.

Reviewer 2 Report

The formulated mathematical model of this paper is plausible. The experimental results also justify the feasibility of using this mathematical model. The computational works of this paper are thorough enough to support authors' arguments. However, the authors are advised to provide the readers an intuitive way to understand why the strength of applied electric field will become dominating factor rather than surface geometry/ roughness at high voltage case.

Author Response

(The authors gave the same response as above.)

Reviewer 3 Report

Dear Authors,

This manuscript investigated the impact of the solid topography and dielectric thickness on electrostatically-induced spreading. The simulation results of continuum-level model were validated with the electrospreading experiments. In addition, the effect of the dielectric thickness on electrospreading was successfully predicted and the dynamics were accelerated with thicker dielectrics.

In overall, this manuscript is very solid and well constructed in terms of theoretical background and experimental validation. Thus, I would recommend this manuscript to be accepted for publication as the current form.

Author Response

(The authors gave the same response as above.)
